# Quantum Dots as a Potential Multifunctional Material for the Enhancement of Clinical Diagnosis Strategies and Cancer Treatments

**DOI:** 10.3390/nano14131088

**Published:** 2024-06-25

**Authors:** Wenqi Guo, Xueru Song, Jiaqi Liu, Wanyi Liu, Xiaoyuan Chu, Zengjie Lei

**Affiliations:** 1Department of Medical Oncology, Nanjing Jinling Hospital, Affiliated Hospital of Medical School, Nanjing University, Nanjing 210000, China; guowenqi0304@163.com (W.G.); 201230020@smail.nju.edu.cn (J.L.); 2Department of Medical Oncology, Jinling Hospital, Nanjing University of Chinese Medicine, Nanjing 210000, China; liuwanyi120@163.com

**Keywords:** quantum dots, tumor diagnosis, tumor-targeted therapy, non-functionalized modified QDs, functionalized QDs

## Abstract

Quantum dots (QDs) represent a class of nanoscale wide bandgap semiconductors, and are primarily composed of metals, lipids, or polymers. Their unique electronic and optical properties, which stem from their wide bandgap characteristics, offer significant advantages for early cancer detection and treatment. Metal QDs have already demonstrated therapeutic potential in early tumor imaging and therapy. However, biological toxicity has led to the development of various non-functionalized QDs, such as carbon QDs (CQDs), graphene QDs (GQDs), black phosphorus QDs (BPQDs) and perovskite quantum dots (PQDs). To meet the diverse needs of clinical cancer treatment, functionalized QDs with an array of modifications (lipid, protein, organic, and inorganic) have been further developed. These advancements combine the unique material properties of QDs with the targeted capabilities of biological therapy to effectively kill tumors through photodynamic therapy, chemotherapy, immunotherapy, and other means. In addition to tumor-specific therapy, the fluorescence quantum yield of QDs has gradually increased with technological progress, enabling their significant application in both in vivo and in vitro imaging. This review delves into the role of QDs in the development and improvement of clinical cancer treatments, emphasizing their wide bandgap semiconductor properties.

## 1. Introduction

Cancer has become a significant threat to human health, with a rising incidence and rising mortality rates. The intricate nature of the tumor microenvironment (TME) and the limitations of conventional treatment methods have highlighted the need for the development of new cancer therapies and enhancements to existing approaches. Quantum dots (QDs), nanoscale semiconductor crystals with unique optical and targeting properties, have demonstrated great potential in cancer diagnoses and treatments. This article commences by presenting the physical properties and cellular uptake pathways of various QDs, such as metal QDs, CQDs, GQDs, BPQDs and PQDs. It then outlines the uses of QDs in cell imaging and in vivo tissue imaging for tumor diagnoses, along with the advancements in functionally modified QDs for tumor treatment. Lastly, the article delves into the challenges and promising future directions of QDs, particularly in terms of their biological toxicity, with the aim of further broadening their applications in the realm of cancer diagnoses and treatments.

## 2. Properties of QDs

### 2.1. Physical Properties of QDs

QDs are commonly semiconductor materials made up of binary and ternary alloys from groups II-VI, III-V, and IV-VI of the periodic table, with dimensions typically ranging from 2 to 20 nm. Their key advantages include (i) their good optical stability and generally long fluorescence lifetimes [1], (ii) the simultaneous excitation of multiple QDs by a single light source [2], and (iii) narrower and tunable emission spectra with broader absorption spectra [3]. The first colloidal QDs, which were made of CdSe, exhibited photoluminescent (PL) properties. In practice, smaller QD particle sizes result in a higher energy, a higher frequency, and bluer emitted light. The low quantum yield (QY) of early CdSe QDs, which was less than 10%, limited their practical applications. Subsequent research efforts have focused on enhancing the QY by coating QD cores with inorganic materials to achieve core-shell structures. This significantly enhances the PL and further increases the QY, as demonstrated by CdSe/ZnS [4], CdSe/CdS [5], and ZnSe/ZnS [6] QDs. Currently, CdSe/CdZnS QDs can achieve a QY of 100%. The continuous optimization of QD properties over the past two decades has greatly advanced their biomedical applications.

Carbon QDs (CQDs), also known as carbon dots or carbon nanodots, are generally smaller than 10 nm and exhibit superior biological properties compared to conventional semiconductor QDs [7]. These properties include a high-water solubility, a strong chemical inertness, easy processing, and excellent photobleaching resistance. The term ‘carbon dots’ was first used in 2006 by Sun et al. when they chemically synthesized fluorescent carbon nanoparticles through the laser ablation of carbon targets and surface passivation [8]. Semiconductor QDs often contain heavy metals, raising concerns about toxicity and limiting their clinical safety. CQDs, on the other hand, overcome this drawback and can be produced at a lower cost, making them a potential replacement for semiconductor QDs.

Graphene QDs (GQDs) differ from CQDs by having an internal graphene lattice, typically less than 100 nm in size and fewer than ten layers thick [9]. GQDs are crystalline zero-dimensional discs derived from two-dimensional graphene. The reduction in the crystal size to a nanoscale size significantly alters the electron distribution, giving rise to quantum confinement GQDs smaller than the exciton Bohr radius [10] and edge effects (energy level shift between intrinsic and edge states, which determines the optical properties [11]). The described effects are a result of significant changes in the electron distribution caused by alterations in the crystal boundaries at the nanoscale level. The photoluminescence mechanism of GQDs is primarily influenced by edge states, quantum confinement, and surface states. In contrast to semi-metallic graphene, GQDs have a non-zero bandgap, making them function as semiconductors or insulators. The opening of the bandgap in GQDs enhances light absorption and elevates the energy spectrum. Similar to CQDs, GQDs contain carboxyl groups that allow for functionalization and surface passivation using a variety of organic, inorganic, and biological materials. Furthermore, GQDs can be functionalized with other heteroatoms or hybridized with different substances to modify their physical structure and chemical properties.

Black phosphorus QDs (BPQDs) have gained significant attention alongside advancements in other two-dimensional materials. Phosphorus is a vital element found in the human body, and exists in three allotropes—red phosphorus, white phosphorus, and black phosphorus. While white phosphorus is chemically unstable and red phosphorus possesses an amorphous structure, black phosphorus stands out as the most stable allotrope, and is known for its non-flammable nature and insolubility in most organic solvents [12]. Black phosphorus is similar to graphene in that it is a layered material consisting of individual stacked atomic layers interacting through van der Waals forces [13]. However, its asymmetric crystal structure, caused by the corrugated pattern of phosphorus atoms, leads to distinct properties compared to graphene: (i) a high mobility and fast carrier relaxation kinetics; (ii) a direct and tunable bandgap in the monolayer and multilayer forms, making it an ideal semiconductor for photoemission and efficient photovoltaic conversion [14]; (iii) a highly anisotropic carrier mobility [15]; and (iv) a photoluminescence burst property [16]. These advantages allow for the development of BPQDs with superior optical properties, showing great potential for applications in bioimaging, cancer therapy, and fluorescence sensing [17].

Metal halide perovskite quantum dots (PQDs), such as CsPbX3 (X = Br, I, and Cl), possess several advantages, including a strong X-ray absorption capability, a high RL intensity, rapid light decay, and a cost-effective solution synthesis, making them a focal point in the field of photocatalytic applications. Examples include light-emitting diodes [18], solar cells [19], fluorescence signal detection, etc. [20]. In addition, compared to classical Cd-based chalcogenide QDs and other QDs, perovskite QDs are of interest because of their significant optical properties, such as their blinking behavior, non-linear absorption, and stimulated emission [21]. X-ray microscopy imaging technology is extensively utilized in the life sciences and other disciplines to accurately visualize the internal microstructure of objects without causing damage. In particular, PQDs play a crucial role in biological imaging due to their exceptional X-ray absorption capabilities.

### 2.2. Common Synthesis Methods of QDs

QDs can be synthesized through physical, chemical, and biological processes. Different synthesis methods will produce QDs with different luminescent properties and biocompatibilities [22]. The synthesis methods of QDs can be categorized into two main types: top-down and bottom-up. Top-down synthesis involves reducing the volume of large bulk semiconductors to create QDs, while bottom-up synthesis typically involves self-assembly. Each of these methods has its own set of advantages and disadvantages (Table 1).

**Table 1 nanomaterials-14-01088-t001:** Common synthesis methods of QDs.

	Method	Advantages	Disadvantages	References
Top-down	Laser ablation	Controllable shape and size	Complex operation and high cost	[8,23]
	Electrochemical oxidation	High purity, high yield, controllable size, and good reproducibility	Complex operation	[24,25,26]
	Chemical oxidation	Easy to operate, large-scale production, no need for sophisticated equipment	Uneven size distribution	[27]
	Ultrasonic treatment	Easy to operate	Instrument waste and high energy costs	[28,29,30]
	Microwave	Shortened reaction time, increased yield and purity	High energy costs	[31,32,33]
	Hydrothermal synthesis	Relatively simple, quick response	Low yield	[34,35,36]
Bottom-up	Cage opening	Strong luminescent properties	Complex operation	[37]
	Thermal decomposition	Simple operation, solvent-free, low cost, and large-scale production	Uneven size distribution	[38]
	Precursor pyrolysis	Stable and strong excitation-dependent photoluminescence	Complex operation and high cost	[39,40,41]
	Ultraviolet irradiation	Mild, clean, and efficient	High energy costs	[42]

### 2.3. Cellular Uptake Modes of QDs

Efficient cellular uptake is a prerequisite for QDs to exert biological effects. The uptake can be broadly categorized into three types, including passive delivery, facilitated delivery, and active delivery (Figure 1 and Figure 2).

Passive delivery depends on the inherent physicochemical properties of the QDs to facilitate endocytosis. In a study by Jaiswal et al., CdSe/ZnS QDs with DHLA were designed and incubated with Hela cells at a concentration of 400–600 nm for 2–3 h. Their findings suggest that the cells took up the QDs through non-specific endocytosis and retained them for several days [43]. Nabiev et al. observed the uptake of CdTe QDs modified with mercaptoacetic acid of different sizes by macrophages, epithelial cells, endothelial cells, etc., and found that 2 nm QDs preferentially appeared on the nucleus and 6 nm QDs preferentially appeared in the cytoplasm of the cells [44]. Kundrotas et al. found in human mesenchymal stem cells that carboxylation-modified QDs were already distributed in the cytoplasm at 1 h, saturated at 6 h, and localized in the perinuclear region at 24 h under conditions of low-density mesenchymal stem cells (Figure 1A) [45]. The uptake of QDs via endocytosis offers a simple method that does not necessitate intricate modifications for cellular uptake. By introducing QDs into the cell system at the correct concentration and incubating for the appropriate duration, cells can uptake QDs through non-specific endocytosis. This allows for the delivery of multiple QDs to cells simultaneously. Nevertheless, the drawbacks of this include the inability to target specific cell types and the heightened cytotoxicity associated with a prolonged exposure to high QD concentrations [46].

The second method involves modifying the surface of QDs with biomolecules or specific chemicals to facilitate the uptake by cells through endocytosis. For example, (i) TAT peptides derived from HIV-1 viruses can be used to modify the surface of QDs, allowing interactions with cell surface receptors. These arginine/lysine-rich TAT peptides, which have a high positive charge, bind to negatively charged receptors on the cell surface (such as acetylheparin sulfate proteoglycans). The TAT peptide can become attached to the dihydrolipoic acid (DHLA)-covered CdSe/ZnS QDs surface by forming coordination bonds between the polyhistidine at the peptide terminus and the Zn atoms on the QD surface. [47,48]. In addition, TAT peptides can become conjugated to SDots to form SDots-TAT. SDots are a novel nanoparticle delivery system that exhibits a strong interaction force with the hydrophobic part of biofilms. They enter the cell mainly through endocytosis (including clathrin-mediated endocytosis and caveolae-mediated endocytosis) (Figure 1B) [49]. (ii) Pep-1 (KETWWWETWWTEWSQPKKKRKV-cysteamine) is an amphiphilic peptide belonging to the cell-penetrating peptide (CPP) family. The QD surface is usually modified with a protein that acts as an intermediate junction, to which Pep-1 binds with a hydrophobic segment, thus promoting cellular uptake. Labeling QDs with Pep-1 would result in a higher labeling efficiency and promote the lysosomal/endosomal escape of QDs [50]. (iii) The modification of QDs by RGD sequences (tripeptide sequences, arginine–glycine–aspartic acid) can facilitate QD delivery. RGD has a specific targeting effect and plays an important role in tumor angiogenesis, proliferation, and metastasis. Thus, RGD-QD can play an important role as a contrast agent in bioimaging [51,52].

The third type is active delivery, i.e., the delivery of QDs by electroporation and microinjection techniques. Bhatia et al. used electroporation to deliver monothiolated PEG-modified CdSe/ZnS QDs into Hela cells [53]. Chen et al. conjugated electroporation with peptide targeting to deliver QDs carrying nuclear-localized peptides into Hela cells via electroporation [54]. Lu et al. further transformed L-particles (nano-oncolytic virus photoparticles) into near-infrared (NIR) fluorescent Ag_2_Se QDs-labeled active tumor-targeting nanocarriers (QDs&DOX@L-particles). Then, they utilized electroporation to instantly form a 1 to 10 nm gap in the cell membrane to successfully deliver the QDs&DOX@L-particles. Through electroporation, Ag_2_Se QDs with DOX were instantly encapsulated into particles with excellent anti-tumor properties. In in vivo experiments, the tumor volume of hormonal mice injected with QDs&DOX@L-particles also showed a significant reduction [55]. Electroporation techniques are suitable for the delivery of large QDs, whereas microinjection techniques allow for the injection of very small volumes of fluid into the cytoplasm. It has been shown that the microinjection of QDs encapsulated in phospholipid block copolymer micelles and microimaging at various stages of embryonic development during the blastomeres stage allows for the distribution of QDs between daughter cells to be observed [56]. Slotkin, J.R et al. injected a mixture of 620 nm QD and farnesylated enhanced green fluorescent protein (EGFP-F) plasmids into the lateral ventricles of mouse embryos using electroporation [57].

**Figure 1 nanomaterials-14-01088-f001:**
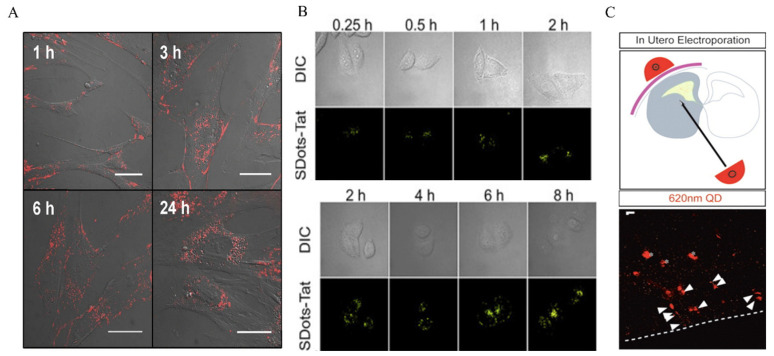
Cellular uptake modes of QDs. (**A**) Passive ingestion: biological effects of intracellular CQD uptake on MSCs. Reproduced from [45], with permission from Kundrotas, G, 2019. (**B**) Facilitated delivery: cell entry by SDots-TAT time-dependent endocytosis. Reproduced from [49], with permission from Dai, J., 2022. (**C**) Active delivery: a mixture of 620 nm QDs and farnesylation-enhanced green fluorescent protein (EGFP-F) plasmids was injected into the lateral ventricle of mouse embryos using electroporation. QDs were found in the ventricular zone (VZ) and subventricular zone (SVZ) 24 h later. Reproduced from [57], with permission from Slotkin, J.R, 2007.

**Figure 2 nanomaterials-14-01088-f002:**
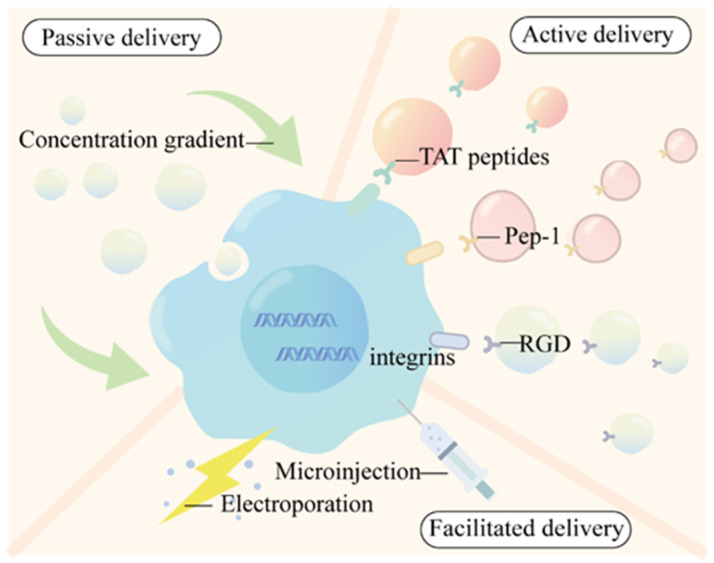
Cellular uptake of QDs.

## 3. The Application of QDs in Tumor Diagnoses

Non-invasive cancer imaging methods have the potential to reduce unnecessary biopsies and improve imaging technology, benefiting clinical treatment. Fluorescence imaging is commonly used for tumor diagnoses, but its clinical applicability is limited due to low visible light transmission in biological tissues. QDs have a higher molar extinction coefficient and absorption rate compared to organic fluorescent dyes, making QD imaging a promising alternative that overcomes the limitations of traditional methods. Additionally, QDs emit light at a faster rate, appearing brighter than organic dyes [58,59,60]. For the optical imaging of deeper tissues, QDs can be utilized within the NIR (700 to 1700 nm) optical range. The field of bioimaging research has seen growth with the incorporation of various types of QDs, such as metal QDs, CQDs, GQDs, BPQDs and PQDs etc., following the initial use of cadmium sulfide QDs in bioimaging applications.

### 3.1. In Vitro Imaging

The application areas of QDs have been substantially expanded by their QY enhancement, particularly in the biomedical disciplines of cell imaging and bioprobes. To create a targeted cancer imaging and sensing system, Lin et al. created unique daunorubicin (DNR)-loaded MUC1 aptamer-NIR CuInS_2_ QD (DNR-MUC1-QD) conjugates. DNR-MUC1-QDs were effectively implemented for prostate cancer cell line imaging [61]. Mona et al. modified CdTe QDs with mercaptosuccinic acid and functionalized CdTe QDs with AS1411 aptamers by the classical EDC-NHS method, and then successfully performed the highly fluorescent and specific molecular imaging of human glioblastoma U87MG cells [62]. In addition, Xu et al. synthesized two types of CdTe/CdS core-shell QDs with different emission wavelengths (545 nm and 600 nm) and obtained the distribution of the fluorescent probes of different QDs according to the different probe colors in the cells to achieve the effect of imaging cancer cells [63]. In addition to the most basic form of imaging, multimodal imaging can provide more than one-dimensional biological information. Multimodal optical imaging refers to the combination of optical imaging with other imaging modalities to obtain multiple parameters at the same time. For example, the combination of magnetic resonance imaging (MRI) technology and optical imaging modalities is a kind of multimodal imaging [64]. Bai et al. constructed MnO_2_/BSA/CdTe QDs that could effectively identify H_2_O_2_ signals in circulating tumors (H_2_O_2_ is a common second messenger in vivo, playing an important regulatory role in cell proliferation, differentiation, and migration) and perform MRI and fluorescence imaging (Figure 3A) [65]. In addition, Rajendra et al. prepared an activatable multimodal/multifunctional composite nanoprobe for direct intracellular imaging and drug delivery using fluorescence-doped Mn-CdS@ZnS QDs modified with superparamagnetic iron oxide nanoparticle (INOP) cores [66]. In addition to QDs, cadmium selenide nanosheets (NPls) compounded with nanorods also contributed to cell imaging. Glioma C6 cells can take up NPls, which exhibit approximately two-orders-of-magnitude-higher 1P and 2P excitation efficiencies in photoluminescence compared to CdSe QDs. This enables detection at significantly lower intracellular NPl concentrations for visualization purposes [67].

Due to their exceptional resistance to photobleaching and biocompatibility, CQDs are particularly well-suited for bioimaging applications, allowing for the precise identification and differentiation of cancer cells. The incorporation of triphenylphosphine salts (TPPs) with CQDs can serve as a targeting ligand for mitochondria. Furthermore, the covalent bonding of borate-protected fluorescein (PFl) to CQDs, acting as an H_2_O_2_ recognition element, enables the monitoring of exogenous H_2_O_2_ levels in L929 cells, and also facilitates the visualization of endogenously produced H_2_O_2_ in pristine 264.7 macrophages [68]. Mn-doped carbon QDs constructed through the coupling of folic acid (FA) and Ce6 are a versatile platform for selectively targeting and detecting cancer cells and exerting PDT effects. In vitro studies have shown that Mn-CQDs@FA/Ce6 can be used as FL/MRI probes for cellular imaging and as photosensitizers for PDT, with laser (671 nm, 1 W cm^−2^) irradiation for 5 min, effectively killing >90% of the cells [69]. Boric acid-functionalized modified CQDs (BNSCQDs) provide better intracellular imaging due to the synergistic effect of nitrogen and sulfur together with boric acid to produce better luminescence properties and due to boric acid binding to the sialyl Lewis receptor, which is over-expressed in the hepatocellular carcinoma cell line HepG2 [70]. Thus, BNSCQD can also specifically target hepatocellular carcinoma imaging [71]. Hyaluronic acid (HA), a natural biopolymer, is known to be a natural ligand for the CD44 receptor, which is known to be commonly overexpressed in many cancer cells. Based on the characteristics of HA, Gao et al. synthesized a therapeutic diagnostic fluorescent nanoprobe, P-CDs/HA-DOX, for self-targeted imaging. P-CDs/HA-DOX enables targeting and penetration into cancer cells. Inside the cell, HAase enzymatically cleaves HA-DOX to release DOX and restore the fluorescence of P-CDs [72]. Nasrin et al. synthesized CQD-FA-HA by further modifying folic acid and HA based on Mg/N-doped CQDs, and CQD-FA-HA was efficiently imaged in both 4T1 and MCF7 cells (Figure 3B) [73]. Zheng et al. synthesized a novel CQD (CD-Asp) using D-glucose and L-aspartic acid as raw materials, demonstrating its ability to target gliomas. In vivo imaging data revealed a high-contrast biodistribution of CD-Asp in a mouse model 15 min post-tail-vein injection. The fluorescence signal in neurogliomas was significantly higher than in normal brain tissues, suggesting the CD-Asp’s capability to traverse the blood–brain barrier and accurately target brain gliomas for effective imaging [74].

GQDs also possess excellent optical properties suitable for bioimaging. At a certain excitation light, GQDs exhibit tunable PL characteristics, which are essential for their bioimaging applications [75]. GQDs have been extensively demonstrated for cellular imaging in Hela cells [76,77]. They have also proven to be effective for cellular imaging in breast cancer cell lines [78,79,80]. Like CQDs, GQDs can be modified by FA and HA for targeted imaging. For instance, Wang et al. utilized FA coupled to GQDs to achieve the highly selective and specific imaging of tumor cells [81]. Notably, GQDs can be utilized for multiphoton fluorescence bioimaging as well. Luo et al. developed adenine-modified GQDs with a high fluorescence, a high QY, and two-photon green fluorescence for two-photon fluorescence cell imaging [82]. Additionally, N-GQD polymer conjugates synthesized by Wu et al. exhibited a high QY, two-photon properties, and a high photoluminescence under acidic pH conditions [83].

BPQDs exhibit good biocompatibility, optical properties, and nonlinear optical properties. However, BP’s high reactivity to water and oxygen can significantly impact the biological properties of BPQDs. Therefore, the stability of BPQDs in an aqueous dispersion can be enhanced for cellular imaging through surface coordination, such as TiL4@BPQDs. Research has demonstrated that TiL4@BPQDs effectively improve the stability of BPQDs in an aqueous dispersion. In MCF7 breast cancer cell line studies, TiL4@BPQDs showed a higher signal sensitivity compared to the commonly used photoacoustic imaging agents (AuNRs), indicating their potential in detecting minute amounts of diseased tissue [84]. Zn-liganded BPQDs were also utilized in this study. Jiang et al. demonstrated that zinc ions can engage in cation–π interactions due to their cationic properties, leading to adsorption on BPs. This suggests that the passivation of the lone pair of electrons of phosphorus can enhance the stability of BPs in both air and water. Zn@BPQDs were found to exert minimal cytotoxicity on 4T1 and Hela cells. Specifically, the 4T1 cells exhibited a strong fluorescence post-endocytosis of Zn@BPQDs, with the fluorescence intensity being dependent on the quantity of Zn@BPQDs internalized by the cells (Figure 3D) [85]. Zhang et al. synthesized BPQDs@DOX@ss-Fe_3_O_4_@C-EGFR NPs using BPQDs as a capping agent, in addition to being a material for combined cancer treatment [86].

PQDs hold significant potential for advancements in biological imaging as a result of their exceptional X-ray absorption capabilities. Nevertheless, the enduring stability of PQDs remains a primary concern. Various studies have indicated that calcium titanium QDs exhibit a high susceptibility to water degradation owing to their intrinsic ionic characteristics [87]. Overcoming the degradation issue of PQDs in polar solvents is a key development priority [88]. Researchers have coated the surface of CsPbBr_3_PQD with dense layers of PS-PEB-PS and PEG-PPG-PEG, resulting in a notable enhancement of the water stability of CsPbBr_3_PQDs without compromising their optical properties. By additionally attaching anti-CD63 antibodies, CsPbBr_3_PQD can be utilized for tracking and imaging exosomes originating from triple-negative MDA-MB-231 breast tumors [89]. Getachew et al. synthesized CsMgxPb1-xI3 QDs and encapsulated them in PF127 Gd (PQD@Gd), enabling dual-mode imaging for fluorescence and MRI. Their experiments demonstrated the internalization of PQD@Gd in HeLa cells via small fossa-mediated endocytosis, leading to successful cellular imaging [90].

**Figure 3 nanomaterials-14-01088-f003:**
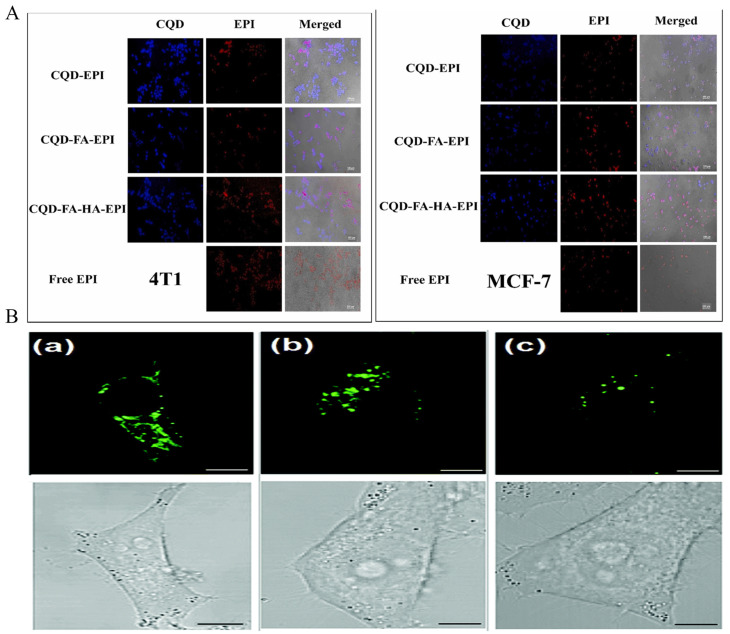
In vitro imaging. (**A**): CQD-FA-HA. Reproduced from [73], with permission from Ziaee, N, 2023 (**B**): confocal (**top**) and bright field (**bottom**) images of HeLa cells labeled with (**a**) ATP GQDs, (**b**) Tr GQDs, and (**c**) excess transferrin molecules and Tr GQDs. Reproduced from [49], with permission from Dai, J, 2022.

### 3.2. In Vivo Imaging

In vivo imaging plays a crucial role in gaining a comprehensive understanding of the mechanisms underlying cancer development and progression in the body. The current methods, such as single-cell labeling using immunohistochemistry or genetics in mice, do not always accurately represent the natural environment. QDs possess unique optical properties that make them ideal for imaging and labeling live animals. As a result, there is growing interest in modifying QDs for in vivo imaging purposes.

A study utilized norbornene-coated QDs (NBPILs) combined with tetrazine-modified antibodies to create QD–norbornene–tetrazine–antibody conjugates (QDNB-TzAb). These conjugates were administered to mice via a tail vein injection, leading to their dispersion throughout the bone marrow and successful labeling the hematopoietic stem and progenitor (Sca1c-kit) cell populations. This research contributed to the exploration of interactions between cells and their microenvironment (Figure 4A) [91]. Han et al. developed an antibody-based, QD-AD, in vivo, single-cell labeling probe that provides a reliable method for studying single-cell interactions and reactions and the microenvironment in living animals [91]. QD-HER2-Ab can be used as an accurate tool for Her2 detection in vivo and can be used for targeted imaging of breast cancer in vivo. [92]. A CdS/chitosan-pAbCD20 QDs is an innovative fluorescent nano-immunocomplex template in non-Hodgkin lymphoma with future potential for tumor detection and imaging [93]. In addition, biocompatible PbS/CdS QDs can be used for non-invasive and high-speed (60 FPS) in vivo imaging by emitting ~1600 nm under an 808 nm excitation light [94]. Wang et al. developed NIR-emitting CdTe/CdS/ZnS core-shell (CSS) QDs. Female nude mice with KB human epidermoid carcinoma tumors were intratumorally injected with CSS780 (λmax: 780 nm) and then imaged using the Maestro in vivo imaging system (CRi Inc.). Remarkably, the tumor tissues treated with CSS780 exhibited strong spectrally resolved signals, highlighting the high sensitivity of in vivo imaging with the synthesized NIR QDs [95]. In addition, PEGylated NIR-IIb QDs allow for the effective penetration of excitation and emission light up to 1.2 mm, which facilitates the effective imaging of the blood flow and tumor vasculature in mice [94]. Similarly, RGD-peptide-functionalized NIR QDs can effectively target αvβ3 integrins in U87MG (human glioblastoma)-loaded mice, allowing for the in vivo and in vitro visualization of the tumor vascular system [96]. Bedia et al. prepared an HA-conjugated nitrogen-doped CQD (HA-CQD) that utilized the HA to achieve tumor-specific targeting along with in vivo imaging capabilities in breast-cancer-loaded mice (Figure 4B) [97].

In addition to common QD materials, Bi_2_O_2_Se QDs have shown some application prospects in biomedical fields in recent years due to the low toxicity and tolerance of their raw materials [98]. Bi_2_O_2_Se QDs were administered intravenously to mice with MCF7 tumors, resulting in the detection of robust photoacoustic imaging signals within the tumors after a specific time. These signals peaked at 8 h post-injection and remained effective for up to 24 h before gradually degrading in vivo. In addition to producing satisfactory imaging signals, the application of 808 nm NIR light irradiation also demonstrated a highly effective tumor-killing capability [99]. The in vivo imaging capability of GQDs has been demonstrated in a variety of model organisms (mouse, zebrafish) [100]. Liu et al. used a simple solvothermal method to prepare N-GQDs as highly efficient two-photon fluorescent probes for imaging cells and deep tissues. The two-photon absorption cross-section of the N-GQDs reaches 48,000 GM, which is higher than that of organic dyes and represents a high value for carbon-based nanomaterials [101]. Additionally, by using the GQD-europium complex (GQD/DBM), it was possible to continue the in vivo fluorescence imaging of Hela-loaded mice, and after 2 h, significant PL signals were observed in the tumor area, with tumor accumulation compared to other organs. This is passive targeting achieved through the EPR (enhanced permeability and retention) effect [102].

## 4. The Potential Application of QDs in Tumor-Targeted Therapy

### 4.1. Non-Functionalized Modified QDs

Non-functionalized QDs serve as the foundation for the creation of different types of QDs. Typically, non-functionalized QDs comprise two key components: QD carrier materials and payloads, which may include drugs or biologically active molecules. These QDs are primarily employed in the chemical drug treatment of tumors, photodynamic therapy, photothermal therapy, immunotherapy, gene therapy, and other related applications (Figure 5 and Table 2).

#### 4.1.1. Chemotherapy

QDs, due to their small particle size, good dispersibility, and excellent biocompatibility, are widely used as drug carrier materials in tumor treatment. It has been reported that QDs can promote the release of various drugs, such as doxorubicin (Figure 6) [103,104,105,106,107,108], paclitaxel [109,110], pemetrexed [111], 5-fluorouracil [112], etc. The use of a single QD carrier in vivo is desirable, as carriers of a moderate size (approximately 10 to 20 nm in diameter) can reduce the reticuloendothelial system uptake and renal clearance rate of drugs, thereby increasing the circulation time in the blood and improving the delivery efficiency [113]. However, to further improve the biocompatibility and reduce long-term toxicity, surface-modified QDs are a good choice for applications in drug delivery systems.

#### 4.1.2. Photothermal and Photodynamic Therapy from Single-Mode to Dual-Mode

Photodynamic therapy (PDT) is an attractive method that involves irradiating the tumor site with an appropriate wavelength, activating photosensitizing drugs that selectively accumulate in the tumor tissue, triggering a photochemical reaction, generating cytotoxic reactive oxygen species (ROS), and thereby killing tumor cells [114]. Compared to traditional anti-cancer therapies, PDT has several advantages. For example, (i) it selectively causes tumor necrosis without harming the surrounding normal tissue. (ii) Photosensitive substances have no special toxicity to the human body, and there is no interaction with other drugs, so they can be used in combination with other therapies. (iii) There is little damage to normal tissues; PDT has a low invasiveness and causes less scarring. It is suitable for superficial lesions and lesions that can be reached by optical fibers. (iv) There is no cumulative toxicity of the drug, and it can be applied repeatedly. It does not require any surgical interventions and it has been clinically used as early as the 1990s, using hematoporphyrin derivatives and photoproteins as photosensitizers for treatment [115]. The practical application of classical photosensitizers is severely limited due to the disadvantages of complex synthesis processes, a poor water solubility, and a short blood circulation time. The rise of nanotechnology has provided new materials for PDT that can be used to overcome the limitations of the first- and second-generation photosensitizers and to develop a third generation of more effective and safer photosensitizers [116]. Good photosensitizers exhibit several key characteristics. Firstly, they maintain stability in compositions. Secondly, they are readily synthesized. Thirdly, they demonstrate non-toxicity in the absence of light exposure. Fourthly, they possess target specificity and can effectively localize to specific tissues, such as tumors. Fifthly, they have a high Φt, with a ΔEt exceeding 94 kJ mol^−1^ = 0.97 eV (the energy of singlet oxygen), and their energy transfer efficiency is adequate for singlet-oxygen formation. Sixthly, they can be efficiently metabolized in the body to minimize noticeable side effects. Seventhly, they prevent or minimize self-aggregation in the body, as aggregation decreases Φt. Lastly, they exhibit an insensitivity to degradation processes, such as photobleaching, to ensure long-term stability [117]. QDs meet the first five standards of good photosensitizers, making them very promising for use as photosensitizers. Only cells or tissues near the QD will be affected by PDT, and QDs have almost no cell toxicity before being activated by PDT, after which any excess unbound QD reagents can be removed from the organism by certain means [118].

In recent years, QDs have been extensively researched as highly promising photosensitizers in the field of PDT. The pioneering use of QDs as photosensitizers was demonstrated in a study by Samia et al., where CdSe QDs acted as the primary energy donor connected to a silicon phthalocyanine photosensitizer through alkyl groups, resulting in emissions at 680 nm. By utilizing Förster resonance energy transfer (FRET) from the QDs to the silicon phthalocyanine photosensitizer, ROS were generated for the purpose of photodynamic cancer therapy [119]. The in vitro treatment of pancreatic cancer cells with CdSe/ZnS QDs under 785 nm NIR laser irradiation notably resulted in observed cell apoptosis (Figure 7A) [120]. Additionally, under 785 nm NIR laser irradiation, MoSe_2_ NDs effectively killed HeLa cells (human cervical cancer cells) at relatively low concentrations (40 μg mL^−1^) [121]. Zhang et al. synthesized copper selenocysteine QDs (Cu-Sec QD) that produced selenocysteine (Sec) radicals, which served not only as reducing agents, but also as capping and stabilizing agents. This effectively addressed the challenges of the poor water solubility and stability observed in conventional nanomaterials, showcasing a favorable biocompatibility. The strong absorption of Cu-Sec QD in the NIR II region (1000 to 1300 nm) facilitated tumor therapy by enhancing the photothermal effects through a Fenton-like reaction, leading to significant cytotoxicity against various cancer cell lines, including human liver, breast, pancreatic, lung, and gastric cancers [122]. Jin-Xuan Fan et al. developed a novel hybrid metal–semiconductor nanocomposite (HNC) using CdSe-seeded/CdS nanorods (NRs). When exposed to visible light in vitro, the HNC produces a higher amount of ROS compared to pure nanorods. Furthermore, by incorporating an RGD modification, the HNC gained targeting capabilities in vivo, leading to its significant accumulation in tumors [123].

CQDs have been widely used in PDT. Novel CQDs synthesized from Diketopyrrolopyrrole (DPP) and CTS have demonstrated an excellent hydrophilicity and biocompatibility. Under 540 nm laser irradiation, they significantly inhibited the growth of tumors in H22 (hepatocellular carcinoma)-tumor-bearing mice [124]. Furthermore, CNQD-CN assembled from g-C3N4-embedded carbon nanosheets [125,126] served as a new type of carbon QD with NIR fluorescence-labeling properties and pH-responsive carrier characteristics, enabling cancer imaging and DOX delivery for cancer cell destruction and providing a multifunctional therapeutic platform for phototherapy and chemotherapy [127]. GQDs under blue light (470 nm) irradiation generate ROS and singlet oxygen, inducing death in U251 human glioma cells through oxidative stress and autophagy [128]. Additionally, innovative methods utilizing oxidative stress have been applied with GQDs to generate excessive ROS, causing G2-/M-phase arrest, cell proliferation inhibition, and apoptosis enhancement to damage colon cancer cells through γ-irradiation beyond NIR radiation [129].

Photothermal therapy (PTT) is a highly effective technique for destroying cancer cells by utilizing PTT agents to induce localized hyperthermia upon exposure to laser light. Unlike traditional cancer treatments such as surgery, chemotherapy, and radiotherapy, PTT reduces harm to surrounding healthy tissues and shows potential as a viable approach for tumor treatment [130,131]. Several photothermal agents have been developed, such as nano-gold, graphene and its derivatives, polydopamine (PDA), indocyanine green, and IR780, which demonstrate a high absorption and efficient photothermal conversion in the NIR region. Nonetheless, these nanomaterials show limited biocompatibility, photostability, and degradability [132,133,134]. MXene is a new type of two-dimensional transition material combining conductivity and hydrophilicity perfectly [135,136,137]. When the two-dimensional size of MXene is less than 100 nm, MXene QDs (MQDs) are formed. The photothermal conversion efficiency of titanium carbide MQDs prepared using a non-fluorination method is significantly higher than that of most previously reported PTT reagents. Additionally, the photostability of MQDs is also excellent. Even after five cycles, the photothermal effect of MQDs does not deteriorate, indicating that they are a potent photothermal agent for inducing apoptosis in cancer cells with promising application prospects [138]. Wang et al. developed nitrogen and boron-doped GQDs (N-B-GQDs) that can effectively convert light energy into heat under external NIR irradiation, achieving PTT and successfully killing cancer cells and inhibiting tumor growth in glioma tumor-bearing mice [139]. Research has shown that biomimetic BPQDs, under NIR irradiation, can effectively kill tumors through photothermal effects. Additionally, they have the ability to promote DC maturation, induce T cell activation, and trigger anti-tumor immune responses. Combining PTT with αPD-L1 immunotherapy not only reprograms the immunosuppressive TME, but also enhances local and systemic anti-tumor immune responses. This combination treatment has been shown to effectively treat distant tumors and inhibit triple-negative breast cancer metastasis through immunologic memory effects (Figure 7B) [140].

**Figure 7 nanomaterials-14-01088-f007:**
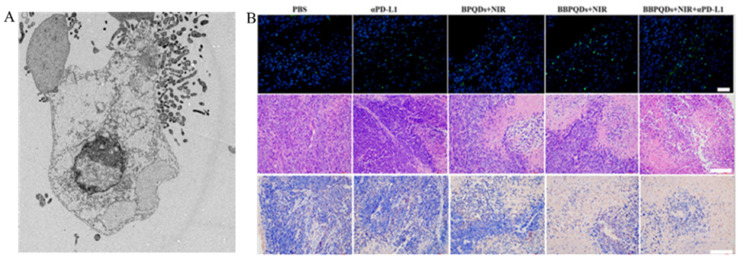
PDT/PTT of non-functionalized QDs: (**A**) CdSe/ZnS + NIR caused significant damage to SW1990 cells. Reproduced from [120], with permission from He, S.J., 2016. (**B**) BBPQDs + NIR induced Ki-67 expression which used H&E and fluorescent-TUNLE-stained in 4T1 tumor tissues. Reproduced from [140], with permission from Zhao, P., 2021.

#### 4.1.3. Other Treatments

Gene therapy is a biological treatment approach that involves transferring normal genes into target cells using gene transfer technology to correct or compensate for diseases caused by gene defects or abnormalities. In the realm of tumor gene therapy, altering the gene expression intensity, inducing tumor necrosis, promoting tumor regression, enhancing anti-cancer responsiveness by modifying genes, repairing target genes to prevent malignant tumor-related thrombosis, and other methods can be employed to treat tumors [141]. While there have been advancements in delivering specific genes through methods such as liposomes and viral vectors, many of these methods have constraints on vector production, leading to increased drug expenses. Moreover, there is a risk that the genetic material transported by viral vectors could integrate close to genes related to cell proliferation, potentially triggering oncogenes [142,143,144]. Therefore, the development of safe and effective non-viral gene vectors is a major challenge in the field of gene therapy.

QDs can target and bind with DNA or mRNA through conjugated oligonucleotide sequences, making them potential non-viral delivery vectors for gene therapy. Current research indicates that QDs are better at inhibiting oncogene activity compared to other methods [145]. For example, GQDs, as gene carriers, can enhance the therapeutic effects of solid tumors by inserting microRNA into the hydrophobic region inside the therapeutic mononuclear cell membrane through the hydrophobic end of the metalloproteinase reaction peptide C18p (Figure 8A) [146]. Cao et al. synthesized a class of porphyrin derivatives (P) known for their high singlet-oxygen-generation capability, along with GQDs that exhibit good fluorescence properties. These were combined with polyethylene glycol (PEG)-functionalized GQDs to create a multifunctional therapeutic diagnostic agent named GQD-PEG-P. GQD-PEG-P is capable of delivering cancer-related miRNA intracellularly, while also demonstrating a high photothermal conversion efficiency of 28.58% and a high QY of 1.08 for singlet-oxygen generation. This dual functionality allows GQD-PEG-P to perform both PDT and PTT, resulting in promising therapeutic efficiency for the combined cancer treatment [147].

In the field of immunotherapy, research has shown that CQDs derived from chlorogenic acid (ChA), a prominent bioactive compound found in coffee, demonstrate notable glutathione peroxidase-like activity. This activity is linked to inducing iron-induced cell death in cancer cells. Furthermore, studies have revealed that ChA CQDs have the ability to attract a significant number of tumor-infiltrating immune cells, such as T cells, NK cells, and macrophages, in mice with liver cancer. This process effectively converts ‘cold’ tumors into ‘hot’ tumors and triggers systemic anti-tumor immune responses [148]. 

In comparison to other tumor treatments, such as gene therapy and immunotherapy, unmodified QDs still have notable limitations. However, QDs that have been functionalized and assembled with specific ligands demonstrate an improved potential for practical applications.

**Table 2 nanomaterials-14-01088-t002:** The application of non-functionalized QDs in tumor-targeted therapy.

Modification of QDs	Modalities of Treatment	QDs	Types of Cancer
Non-functionalized modified QDs	Chemotherapy	QDs carry doxorubicin [103,104,105,106,107,108]	Breast cancer Lung cancer Cervical cancer Hepatocarcinoma Glioblastoma
		QDs carry paclitaxel [109,110]	Hepatocarcinoma Prostate cancer
		QDs carry pemetrexed [111]	Breast cancer
		QDs carry 5-fluorouracil [112]	Lung cancer
	PTT/PDT	CdSe/ZnS QDs [120]	Pancreatic cancer
		MoSe2 NDs [121]	Cervical cancer
		Cu-Sec QDs [122]	Hepatocarcinoma
		DPP-CTS-CQDs [124]	Hepatocarcinoma
		CNQD-CN [127]	Cervical cancer
		GQDs [129]	Glioblastoma Colon cancer
		MXene-MOD [138]	Cervical cancer
		N-B-GQDs [139]	Glioblastoma
		BBPQDs [140]	Breast cancer
	Other treatments	C18p-GQDs [146]	Breast cancer
		GQD-PEG-P [147]	Lung cancer
		ChA CQDs [148]	Hepatocarcinoma

### 4.2. Functionalized QDs

#### 4.2.1. Lipid and Polysaccharide Modification

Lipids, including fats (triglycerides) and lipids (phospholipids, sterols), are compounds formed from the interaction of fatty acids and alcohols. These components are crucial for cell membranes, cellular recognition, and signal transduction. Exosomes (EXOs) are lipid membrane vesicles (50 to 150 nm) secreted by multivesicular bodies through endocytosis, and they offer advantages such as easy accessibility, an excellent immunoevasive ability, and extended circulation time in the blood. Notably, EXOs can be efficiently taken up by cells and directed towards specific tissues or cells based on biological modifications on their membrane. These characteristics make EXOs a promising carrier for modifying QDs. Research has used electroporation to package BPQDs into EXOs, creating BPQDs@EXO nanospheres (BEs) to evaluate their penetration and cytotoxic capabilities in bladder-cancer-like tissues. The results showed that the optimal concentrations of BEs demonstrated significant penetration into tumor tissues, with the corresponding photothermal effect effectively inhibiting tumor growth and angiogenesis. In vivo studies on mice with subcutaneous bladder cancer showed similar outcomes [149]. Polysaccharides, also known as glycans, are high-molecular-weight compounds formed by numerous monosaccharides through glycosidic bonds. Similar to lipids, they are crucial components of cell membranes and play vital roles in cellular recognition and signal transduction. Among polysaccharides, HA stands out due to its specific binding to CD44, making it suitable for targeting tumor cells. Consequently, HA has been employed in cancer-targeted therapy. In a study by Nafiujjaman et al., a PDT system was developed by combining Ce6 with HA and loading it onto GQDs. This system demonstrated outstanding biocompatibility and PDT effects when exposed to light irradiation [150]. I have summarized in Table 3.

#### 4.2.2. Protein Modification

The functionalized modification of QDs by proteins was documented as early as 2003. In one study, semiconductor CdSe QDs were linked to Pc4 (a human positive coactivator known as a PDT drug), leading to a significant expansion of the excitation light wavelengths for Pc4 through fluorescence resonance energy transfer (FRET). It can be estimated that the FRET efficiency can reach 77%. This innovative pairing of QDs with Pc4 offers the advantage of being able to activate photosensitizer molecules with varying excitation light wavelengths. Additionally, the spectral properties of the QDs can be fine-tuned to match those of any PDT photosensitizer simply by adjusting the size of the QDs [119]. Peptide E5-modified CdSe/ZnS QDs significantly enhanced the binding affinity to CXCL4-overexpressing cancer cells and inhibited the CXCL12-induced migration of cancer cells [151]. L-cysteine-capped CdSe QDs coupled with methotrexate (MTX) efficiently circumvented the efflux of drug-resistant cancer (KB) cells through the cellular endocytosis of QDs on KB cells (MTX-resistant cells), with a significant killing effect [152]. IL-13-modified CdSe QDs can detect tumor exosomes and serve as ex vivo markers of glioma stem cells and exosomes, providing information for the diagnosis and prognosis of patients with malignant diseases [153]. According to the specific antigen–antibody recognition mechanism, QDs can also be modified by antibodies to achieve specific targeting for cancer cells. Several studies have demonstrated that the GP73 protein is upregulated in hepatocellular carcinoma and can be used as a prognostic marker [154]. Liu et al. covalently linked Mn-modified CdTe/CdS QDs with GP73 antibodies using EDC as a cross-linking agent. Subsequently, they specifically attached protein A/G agarose beads to the antibody bound to the QDs. These QD–Ab–bead bioconjugates can selectively capture GP73 in the samples through the immunoreactivity between GP73 and the GP73 antibody. The immune complexes can be easily separated by centrifugation. The fluorescence intensity of the QD–Ab–microbead biosensor is reduced upon binding with GP73, enabling the direct detection of GP73 by monitoring the fluorescence intensity change in the QD–Ab beads when they bind to specific target analytes [155]. 

The use of PDT/PTT with QDs for killing cancer cells is highly effective, especially when combined with chemotherapeutic agents. BPQDs are particularly promising due to their large NIR extinction coefficient, high photothermal conversion efficiency, and low cytotoxicity, making them ideal photosensitizers for tumor treatments. Zhang et al. utilized EGFR for tumor targeting in the synthesis of BPQDs@DOX@ss-Fe_3_O_4_@C-EGFR NPs, where BPQDs and Fe_3_O_4_@C NPs served as photothermal agents. Upon the NIR irradiation of A549 and MCF7 tumor-bearing mice, a significant PTT effect was observed due to a rapid increase in the tumor temperature, thus enhancing the efficacy of tumor killing through a combination of chemotherapy and PTT with DOX release [86]. I have summarized in Table 3.

#### 4.2.3. Organic Polymer Modification

The folate receptor, which is overexpressed on the surface of cancer cells, is a very attractive therapeutic diagnostic target [156]. CdSe-aza-BODIPY QDs coated with folic acid and polyethylene glycol induced ROS upon irradiation at 653 nm in Hela cells. The QDs were found to be preferentially localized in cancer cells, with negligible localization of the QDs in normal cells. This demonstrated the specificity, efficacy, and limited side effects of the QDs [157]. Li et al. developed folate-conjugated CdTe/CdS chiral QDs (FA-Cys-CdTe/CdS) by combining two chiral ligands, folate and cysteine. The chiral nature of the FA-Cys-CdTe/CdS was evidenced by the CD signals below 250 nm for folate and at 270 nm for cysteine-CdTe/CdS. This study revealed that FA-Cys-CdTe/CdS exhibited a notable inhibitory effect on cancer cell growth while showing minimal impact on normal cells. Additionally, the QDs were observed to target cancer cells and induce apoptosis, suggesting their potential as anti-tumor agents [158]. Rajendra et al. functionalized superparamagnetic iron oxide nanoparticle cores (INOPs) with fluorescent-doped Mn CdS@ZnS QDs. The resulting INOP-Mn CdS@ZnS can effectively target and eliminate cancer cells. Their study incorporated STAT3 inhibitors, folic acid for cancer cell targeting, and mPEG for hydrophilicity to enhance the QDs. This led to the fluorescence recovery of the QDs upon their uptake by breast cancer cells in glutathione-rich environments, reducing the viability of MDA-MB-231 and Panc-1 cell lines. Notably, the viability of mouse thymus basal cells (TE-71 cells) remained unaffected. Overall, this study significantly enhanced the delivery efficiency of anti-cancer drugs, improving the therapeutic efficacy while minimizing the side effects from excess drug presence [66]. 

In addition to the core-shell QDs, pH-responsive D-biotin/DOX-loaded mPEG-OAL/N-CQDs possess the ability to remain stable at a physiological pH and release DOX in acidic environments. This enables DOX to be effectively absorbed by cancer cells through D-biotin-mediated cellular endocytosis, allowing it to act in the nuclear region and exert pharmacological effects [159]. UCNP nanoparticles modified with GQDs (UCNP-GQDs) can be activated by NIR light to emit energy from the UCNP, subsequently exciting the GQDs and generating singlet oxygen (O^−1^). Researchers have also covalently attached TRITC to UCNP-GQDs, imparting the material with precise targeting capabilities towards mitochondria. This targeted approach can induce irreversible apoptosis in tumor cells within mitochondria, leading to a notable enhancement in the effectiveness of PDT [160].

Tumor-specific targeting can be achieved through passive targeting by utilizing Poly PLGA loaded with 3 nm BPQDs processed via the solvent evaporation of oil-in-water emulsions to create BPQD/PLGA nanospheres of approximately 100 nm in size. Nanomaterials ranging from 20 to 200 nm can effectively evade renal filtration, resulting in the passive accumulation of BPQD/PLGA nanospheres at the tumor site through the enhanced permeability and retention (EPR) effect. Following the injection of Cy5.5-labeled BPQD/PLGA nanospheres in MCF7-loaded mice, strong fluorescence was still detectable after 48 h. A subsequent fluorescence analysis of tumor tissue sections revealed a widespread distribution of nanospheres throughout the entire tumor section, indicating the successful penetration of the BPQD/PLGA nanospheres. Furthermore, by harnessing the photothermal properties of the BPQD/PLGA nanospheres for PTT with 808 nm NIR irradiation, the tumor site was rapidly heated to a peak temperature of 58.8 °C, effectively inducing tumor ablation. In glioma C6 and breast cancer MCF7 cells, NIR irradiation successfully enhanced the PDT effect of BPQDs, resulting in significant cell death. Therefore, BPQD/PLGA nanospheres exhibit excellent passive targeting capabilities and serve as effective photothermal agents [161]. Further PEG modifications significantly enhanced the biostability of the BPQDs [17]. Li et al. selected three organic polymers to modify BPQDs, resulting in BPQD@PAA/PEG/PPS. The self-assembly of BPQD@PAA/PEG/PPS led to the embedding of Ag in the PPS shell layer, leveraging the hydrophobicity of the PPS shell for the effective protection of Ag ions and the acquisition of BP Ve-Ag. Upon an intravenous injection of BP Ve-Ag in 4T1-bearing mice, the BP Ve-Ag QDs exhibited specific accumulation at the tumor site over time. A subsequent 660 nm irradiation induced significant tumor cell apoptosis and necrosis. The TME showed elevated levels of pro-inflammatory factors, which could potentially enhance anti-cancer immunotherapy [162]. In addition to cancer-killing through PDT and PTT, BPQDs have also shown promise as a gene delivery system. When functionalized with polyelectrolyte polymers, BPQDs can effectively deliver lysine-specific demethylase and siRNA to PA-1 cells (human ovarian teratocarcinoma). When combined with NIR light, this delivery system can inhibit cancer growth by more than 80%. These findings highlight the potential of BPQDs for siRNA delivery and their synergistic anti-tumor effects when combined with photothermal PDT [163]. I have summarized in Table 3.

#### 4.2.4. Other Modifications

Guo et al. conducted a study where they performed amino-functionalized modifications to GQDs to create amino-N-GQDs. These modified GQDs were found to serve as effective two-photon probes that showed a strong PDT effect by producing a significant amount of ROS and effectively targeting multidrug-resistant substances [139]. Most BPQD-mediated photothermal diagnostics are typically constrained by limited tissue penetration, resulting in minimal competition in the NIR-II and X-ray-induced phototherapy domains. However, Li et al. successfully synthesized Nd^3+^-coordinated BPQDs, thus forming ion-coordinated BPQDs (BPNd), which significantly enhanced their competitiveness in the realms of NIR-II and X-ray-induced phototherapy. This enhancement was attributed to the presence of recyclable NIR/X-ray photoelectric switching effect ions between the BPQDs and Nd^3+^. Furthermore, BPNd exhibits the ability to effectively traverse the blood–brain barrier for intracranial fluorescence imaging, enabling the detection of glioblastoma growth and the inhibition of its progression through X-ray-induced synergistic photodynamic chemotherapy [164]. I have summarized in Table 3.

**Table 3 nanomaterials-14-01088-t003:** The application of functionalized QDs in tumor-targeted therapy.

Modification of QDs	Type of Modification	QDs	Types of Cancer	Treatment
Functionalized QDs	Lipid and polysaccharide modification	BPQDs@EXO [149]	Bladder cancer	PTT
		Ce6-HA-GQDs [150]	Non-small-cell carcinoma	PDT
	Protein modification	Peptide E5-modified CdSe/ZnS QDs [151]	Cervical cancer	Ligand-receptor-specific binding
		L-cysteine-capped CdSe QDs coupled with methotrexate (MTX) [152]	Oral epidermal carcinoma	Chemotherapy
		IL-13-modified CdSe QDs [153]	Glioma	Ligand-receptor-specific binding
		CdTe/CdS QDs with GP73 [155]	Hepatocarcinoma	Specific binding of antigen to antibody
		BPQDs@DOX@ss-Fe3O4@C-EGFR NPs [86]	Breast cancer	PDT
	Organic polymer modification	CdSe-aza-BODIPY QDs [157]	Cervical cancer	PDT
		FA-Cys-CdTe/CdS [158]	Breast cancer	Ligand-receptor-specific binding
		INOP-Mn CdS@ZnS [66]	Breast cancer	Chemotherapy
		mPEG-OAL/N-CQDs [159]	Cervical cancer	Chemotherapy
		TRITC-UCNP-GQDs [160]	Breast cancer	PDT
		BPQDs/PLGA NS [161]	Breast cancer glioma	PTT
		BP Ve-Ag QDs [162]	Breast cancer	PDT
		siRNA-BPQDs [163]	Ovarian teratocarcinoma	PDT
	Other modification	amino-N-GQDs [139]	Oral epidermal carcinoma	PDT
		BPNd [164]	Glioblastoma	PDT

## 5. Challenges and Summary

QDs represent a new generation of nanomaterials that have garnered significant interest in the biomedical field due to their customizable optical properties and high tissue compatibility. Since their introduction, a variety of QDs with distinct advantages have been developed to meet various clinical needs in biomedicine. Starting from the initial CdSe QDs, this field has seen the gradual emergence of CQDs, GQDs, BPQDs and PQDs. These advancements have the potential to revolutionize cancer treatment and early detection imaging. The anti-tumor effects of QDs primarily involve PDT and PTT, with many QDs capable of inducing ROS through the generation of singlet oxygen via triplet energy transfer (TET) and enhancing the PL of conventional photosensitizers through FRET. Furthermore, QDs can be further functionalized with common modifications such as polysaccharides, lipids, proteins, organic polymers, and other modifications to achieve specific tumor targeting for effective killing and cellular imaging applications, thereby minimizing undesired side effects.

Fewer studies have focused on exploring the long-term toxicity of different QDs on living organisms. The toxicity of QDs is primarily determined by factors that include the chemical composition, particle size, surface modifications, exposure pathways, duration of exposure, and dosage [165]. (i) The toxicity caused by their chemical composition is mainly due to the heavy metal core of metal QDs. Such QDs may be ingested into the cytoplasm through cellular endocytosis, and their prolonged presence in the cellular environment may damage the outer coating of the QDs, thus releasing the heavy metal core of the QDs [166]. (ii) The surface reactivity of nanoscale QDs raises toxicological concerns, which are often addressed through surface passivation techniques such as the use of polyethylene glycol. QDs are semiconductor materials and are typically smaller than 20 nm, making them easily absorbed by the human body and distributed in various organs. Research indicates that larger QDs tend to exhibit higher toxicity levels [167] (accumulation in spleen). (iii) Surface modifications are typically implemented to enhance biocompatibility; however, improper modification techniques can introduce unknown toxicity. Rashi et al. studied the toxicity of CdSe QDs with various surface modifications on human bronchial epithelial cells. Their research revealed that the charge of the QDs has a more significant effect on toxicity compared to the size of functionalization [168]. (iv) Normally, QDs are injected intravenously into the body, which can potentially harm blood vessels [169]. But there are also studies indicating that, due to the small size of QDs, they can also enter the body through the skin and respiratory tract [170]. (v) The toxicity of QDs is intricately linked to the dosage and duration of exposure. Higher concentrations and longer exposure times of QDs tend to increase their biological toxicity. In a study by Wang et al., it was observed that exposing mouse livers and kidneys to CdTe QDs led to a dose-dependent elevation in the activities of superoxide dismutase (SOD), catalase (CAT), and glutathione peroxidase (GPx) [171]. In addition to considering the biological toxicity of QDs, it is crucial to assess their potential accumulation in the body over time and how the human body eliminates these nanoparticles. Some studies suggest that QDs with a diameter below 5.5 nm can be excreted through urine, while those larger than 8 nm tend to persist in the bloodstream [172]. Ge et al. also found that AgSe QDs (<3 nm) could be removed from mouse kidneys after 168 h without long-term accumulation [173]. In addition to renal clearance, QDs can bind to serum proteins and be excreted through the liver [174]. The biological toxicity of QDs is influenced by various factors, making it necessary for the biological safety assessment of QDs to consider multiple aspects rather than simply labeling them as ‘toxic’ or ‘non-toxic’. More clinical data are required to understand the metabolism and excretion of QDs in humans. The introduction of new inorganic QDs such as CQDs, GQDs, BPQDs, and PQDs is expected to reduce the biological toxicity associated with older QDs. For instance, studies have demonstrated that GQDs exhibit a low toxicity, and further surface modifications can enhance their therapeutic effects while ensuring long-term stability during biological therapy. Novel nanomaterials such as BPQDs show promise in lowering the biotoxicity risks due to phosphorus being a natural element in the human body. These advancements aim to enhance the targeting efficiency of QDs, minimize off-target effects, improve QD accumulation in tumor tissues, and diminish the impacts on normal tissues. The ongoing advancement of QDs holds promise for creating materials with superior optical properties, a reduced toxicity, and enhanced biocompatibility. These advancements have the potential to enhance the efficacy of various QD-based therapies in cancer treatment, including loading chemotherapy drugs onto QDs, utilizing QDs for PTT and PDT, and exploring gene therapies. Furthermore, the continued development of QDs is expected to lead to the creation of improved imaging agents, bolster bioimaging technology, and enable the precise in vivo imaging of cancer cells with deep tissue penetration and minimal background fluorescence.

## Figures and Tables

**Figure 4 nanomaterials-14-01088-f004:**
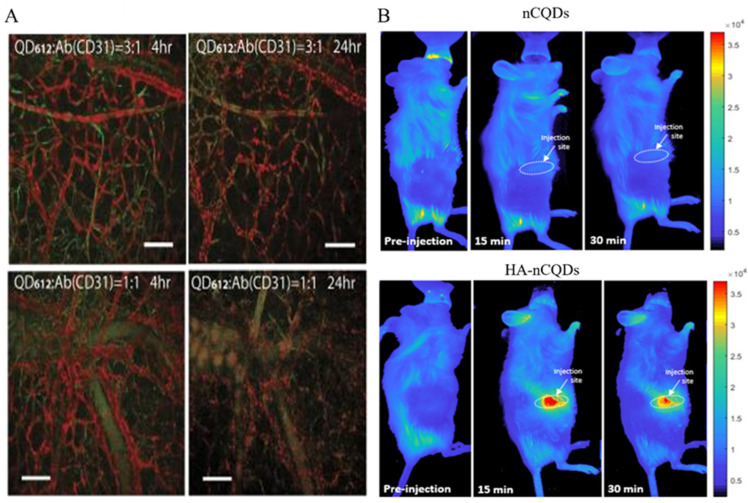
In vivo imaging: (**A**) QD-Ab. Reproduced from [91], with permission from publisher Han, H.S, 2015. (**B**) nCQDs and HA-nCQDs. Reproduced from [97], with permission from Karakocak, B.B., 2021.

**Figure 5 nanomaterials-14-01088-f005:**
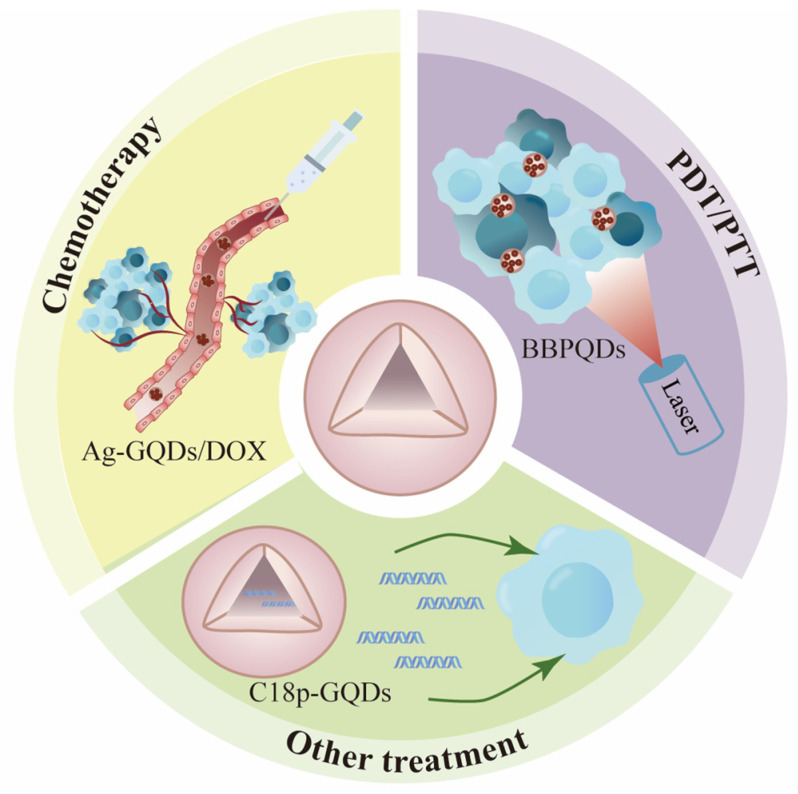
The application of non-functional QDs in tumor therapy.

**Figure 6 nanomaterials-14-01088-f006:**
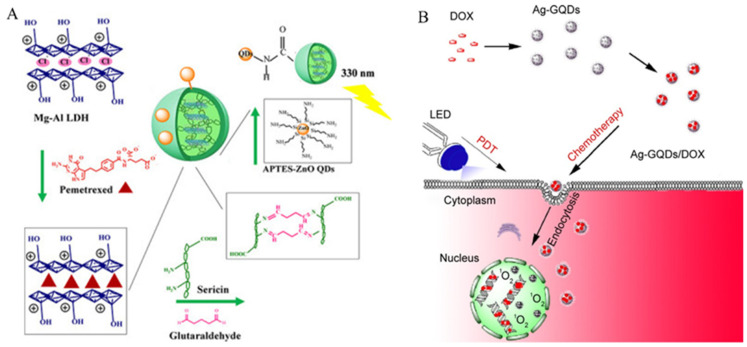
Chemotherapy of non-functionalized QDs: (**A**) APTES-ZnO QDs/Seri@LDH2-PMX. Reproduced from [103], with permission from Abdelgalil, R.M, 2023. (**B**) Ag-GQDs/DOX. Reproduced from [105], with permission from Habiba, K., 2016.

**Figure 8 nanomaterials-14-01088-f008:**
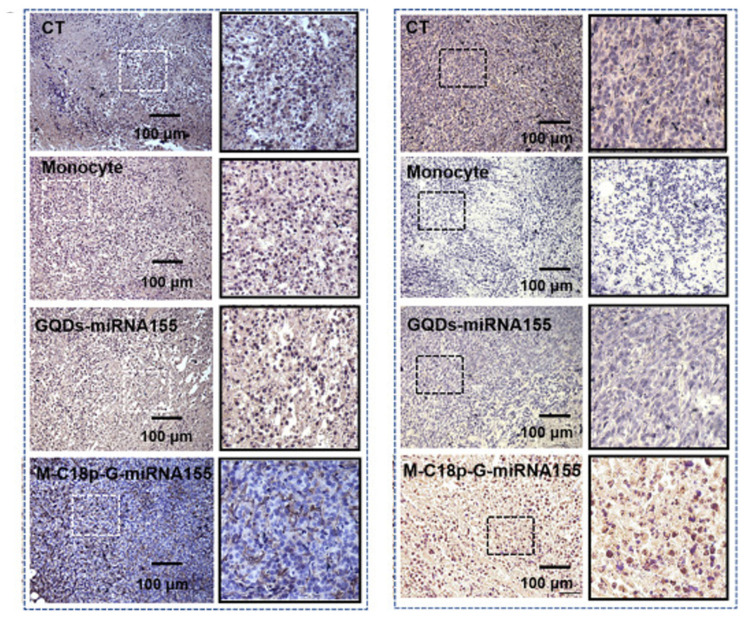
Other treatments of non-functionalized QDs. After the QD treatment, the tumor sections were stained with Ki67 and TUNEL, respectively. Reproduced from [146], with permission from Xia, Q., 2023.

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
