# Peer review of "Quantum Dots as a Potential Multifunctional Material for the Enhancement of Clinical Diagnosis Strategies and Cancer Treatments"

_nanomaterials, 2024, doi:10.3390/nano14131088_

Round 1
Reviewer 1 Report
Comments and Suggestions for Authors
The manuscript is well-written and, when published, will help readers improve and arrange their understanding of the science of quantum dots and related species. However, I need to suggest a few remarks.
1. Different authors use the terms "carbon dots" or "carbon quantum dots" either as synonymous and consider carbon dots as belonging to the more general class of quantum dots, or insist that carbon dots are not quantum dots because they don't demonstrate quantum confinement; therefore, it would be very helpful to clarify the ambiguity in the review. I would also suggest considering some papers on carbon dots published by Prof. Rogach of City University of Hong Kong (the reviewer is in no relationship with him).
2. Line 44: "QDs are typically semiconductor materials composed of binary alloys..."; I do believe that ternary I-III-VI quantum dots, less toxic promising alternative to the binary QDs, should also be discussed in some detail in the review. Also, it might be good to say a few words about such relatives of semiconductor quantum dots as nanorods and nanoplatelets.
3. I am wondering why Perovskite quantum dots were not considered in the review; I do believe they deserve proper mention in the context.
In line 630, "CeSd" should probably be replaced by "CdSe".
The statement on biocompatibility of (binary) quantum dots may be argued by many researchers, it should be commented.
Author Response
Thank you for reviewing our manuscript and providing constructive feedback, which greatly assisted in enhancing the clarity and flow of the review. We have carefully revised the manuscript and addressed each point in the response provided below. The revised sections are highlighted in yellow in the manuscript, while our responses are presented in blue text.
Comment 1: Different authors use the terms "carbon dots" or "carbon quantum dots" either as synonymous and consider carbon dots as belonging to the more general class of quantum dots, or insist that carbon dots are not quantum dots because they don't demonstrate quantum confinement; therefore, it would be very helpful to clarify the ambiguity in the review. I would also suggest considering some papers on carbon dots published by Prof. Rogach of City University of Hong Kong (the reviewer is in no relationship with him).
Reply 1: The article discussing Prof. Rogach of City University of Hong Kong delves into optoelectronic applications of materials and the synthesis of new materials. While I initially found it not directly relevant to my article, it led me to explore other related articles on CQDs and enriched my writing content further. An article by Wang and YF in 2014 clearly defined carbon dots and carbon quantum dots as a kind of substance. In the 2014 article, the author explained that Carbon-based quantum dots consisting of graphene quantum dots (QGDs) and carbon quantum dots (CQDs, C-dots or CDs) are a new class of carbon nanomaterials with sizes below 10 nm. They were first obtained during the purification of single-walled carbon nanotubes through preparative electrophoresis in 2004, and then via laser ablation of graphite powder and cement in 2006. The references about 2004 and 2006 are as follows::doi: 10.1021/ja040082h.(2004);10.1021/ja062677d(2006).
I have also appropriately altered the narrative of this paragraph regarding CQDs. Carbon quantum dots (CQDs), also known as carbon dots or carbon nanodots, are generally smaller than 10 nm and exhibit superior biological properties compared to conventional semiconductor QDs[7]. These properties include high water solubility, strong chemical inertness, easy processing, and excellent photobleaching resistance. The term 'carbon dots' was first used in 2006 by Sun et al. when they chemically synthesized fluorescent carbon nanoparticles through laser ablation of carbon targets and surface passivation.[8]. Semiconductor QDs often contain heavy metals, raising concerns about toxicity and limiting their clinical safety. CQDs, on the other hand, over-come this drawback and can be produced at a lower cost, making them a potential re-placement for semiconductor QDs.
Changes in the text: We have modified our text as advised (see Page 2, Line 59-68).
Comment 2: Line 44: "QDs are typically semiconductor materials composed of binary alloys..."; I do believe that ternary I-III-VI quantum dots, less toxic promising alternative to the binary QDs, should also be discussed in some detail in the review. Also, it might be good to say a few words about such relatives of semiconductor quantum dots as nanorods and nanoplatelets.
Reply 2: First, thank you very much for your suggestion. There are few articles on biological applications of I-III-VI quantum dots. Most of them focus on applications in the field of optoelectronics. Therefore, it was not added to the text. Secondly, adding content about semiconductor quantum dots as nanorods and nanoplatelets will indeed further enrich the readability of the article. Relevant content has been added to the text. For example, Jin-Xuan Fan et al. developed a novel hybrid metal–semiconductor nanocomposite (HNC) using CdSe-seeded/CdS nanorods (NRs). When exposed to visible light in vitro, HNC produces a higher amount of reactive oxygen species (ROS) compared to pure nanorods. Furthermore, by incorporating RGD modification, HNC gains targeting capabilities in vivo, leading to significant accumulation in tumors. cadmium selenide nanosheets (NPls) compounded with nanorods also contribute to cell imaging. Glioma C6 cells can take up NPls, which exhibit ap-proximately two orders of magnitude higher 1P and 2P excitation efficiency in photoluminescence compared to CdSe QDs. This enables detection at significantly lower in-tracellular NPl concentrations for visualization purposes.
Changes in the text: We have modified our text as advised (see Page 5, Lines 277-232 and Page 12, Lines 435-439)
Comment 3: I am wondering why Perovskite quantum dots were not considered in the review; I do believe they deserve proper mention in the context.
In line 630, "CeSd" should probably be replaced by "CdSe".
Reply 3: Thank you very much for your suggestions. I did ignore PQDs before, and now I have added relevant content about PQDs to the article. The unique properties and advantages of PQDs were introduced, and research on in vitro imaging of PQDs was also written to a certain extent.
Metal halide perovskite quantum dots (PQDs), like CsPbX3 (X=Br, I, and Cl), possess several advantages including strong X-ray absorption capability, high RL intensity, rapid light decay, and cost-effective solution synthesis, making them a focal point in the field of photocatalytic applications. For example, light-emitting diodes [18]、solar cell [19]、Fluorescence signal detection, etc [20]. In addition, compared to classical Cd-based chalcogenide QDs and other QDs, perovskite QDs are of interest because of their significant optical properties, such as blinking behavior, non-linear absorption, and stimulated emission[21]. X-ray microscopy imaging technology is extensively utilized in the life sciences and other disciplines to accurately visualize the internal microstructure of objects without causing damage. Particularly, PDQs play a crucial role in biological imaging due to their exceptional X-ray absorption capabilities.
Phosphorus quantum dots (PQDs) hold significant potential for advancement in bio-logical imaging as a result of their exceptional X-ray absorption capabilities. Nevertheless, the enduring stability of PQDs remains a primary concern. Various studies have indicated that calcium titanium QDs exhibit high susceptibility to water degradation owing to their intrinsic ionic characteristics. [87]. Overcoming the degradation issue of PQDs in polar solvents is a key development priority [88]. Researchers have coated the surface of CsPbBr3PQD with dense layers of PS-PEB-PS and PEG-PPG-PEG, resulting in a notable enhancement of the water stability of CsPbBr3PQD without compromising its optical properties. By additionally attaching anti-CD63 antibodies, CsPbBr3PQD can be utilized for tracking and imaging exosomes originating from triple negative MDA-MB-231 breast tumors[89]. Getachew et al. synthesized CsMgxPb1-xI3 quantum dots and encapsulated them in PF127 Gd (PQD@Gd), enabling dual-mode imaging for fluorescence and MRI. Their experiments demonstrated internalization of PQD@Gd in HeLa cells via small fossa-mediated endocytosis, leading to successful cellular imaging[90].
Changes in the text: We have modified our text as advised (see Page 3, Line 101-111 and see Page 7, Line 293-306)
I'm very sorry for the clerical error in my article. I have corrected it.
Changes in the text: We have modified our text as advised (see Page 18, Line 678)
Reviewer 2 Report
Comments and Suggestions for Authors
In their review titled, "Quantum Dot as a Potential Multifunctional Biomaterial Enhances the Clinical Diagnostic Ability and Therapeutic Effect of Cancer", Wenqi Guo et al., provide a well-organized recapitulation of the roles of quantum dots in strategies aimed at improving clinical cancer diagnosis and treatment. The authors introduce the physical properties and cellular pathways for QD uptake, then summarize applications for diagnosis and treatment. They finally discuss challenges and provide promising future directions. I suggest the following minor changes/corrections.
1. The title sounds misleading. QD do not enhance the diagnostic ability of cancer. QDs do not enhance the therapeutic effect of cancer. Rather, QDs enhance strategies/procedures/modalities for the diagnosis/treatment of cancer. Hence, the titled could now be: "Quantum Dot as a Potential Multifunctional Biomaterial for the Enhancement of Strategies for the Clinical Diagnosis and Treatment of Cancer", or something similar/clearer.
2. The statement in line 513 (that is, earliest report of functionalization in 2003) needs immediate citation.
3. It seems the statement in lines 635-639 is not a complete sentence.
The English language is fairly good and comprehensible. Minor but careful redaction of sentences to provide directness and enhance clarity is strongly recommended.
Author Response
Thank you for reviewing our manuscript and providing constructive feedback, which greatly assisted in enhancing the clarity and flow of the review. We have carefully revised the manuscript and addressed each point in the response provided below. The revised sections are highlighted in yellow in the manuscript, while our responses are presented in blue text.
Comment 1: The title sounds misleading. QD do not enhance the diagnostic ability of cancer. QDs do not enhance the therapeutic effect of cancer. Rather, QDs enhance strategies/procedures/modalities for the diagnosis/treatment of cancer. Hence, the titled could now be: "Quantum Dot as a Potential Multifunctional Biomaterial for the Enhancement of Strategies for the Clinical Diagnosis and Treatment of Cancer", or something similar/clearer.
Reply 1: Thank you very much for your suggestion. The title of the article has been changed to “Quantum Dot as a Potential Multifunctional material for the Enhancement of Strategies for the Clinical Diagnosis and Treatment of Cancer”
Changes in the text: We have modified our text as advised (see Page 1, Lines 1-3)
Comment 2: The statement in line 513 (that is, earliest report of functionalization in 2003) needs immediate citation.
Reply 2: Thank you for carefully reviewing my article. The reference to this article in 2003 is just after this sentence. The later CdSe were linked to Pc4 is the QDs in the 2003 article. Functionalized modification of QDs by proteins was documented as early as 2003. In one study, semiconductor QDs CdSe were linked to Pc4 (a human positive coactivator known as a photodynamic therapy drug), leading to a significant expansion of the excitation light wavelengths for Pc4 through fluorescence resonance energy transfer (FRET efficiency reaching up to 77%). This innovative pairing of QDs with Pc4 offers the advantage of being able to activate photosensitizer molecules with varying excitation light wavelengths. Additionally, the spectral properties of the QDs can be fine-tuned to match those of any photodynamic therapy photosensitizer simply by adjusting the size of the QDs [119].
Changes in the text: We have modified our text as advised (see Page 15, Lines 555-562)
Comment 3: It seems the statement in lines 635-639 is not a complete sentence.
Reply 3: I am very sorry that this paragraph was written in error and have changed the narrative of this paragraph.
QDs represent a new generation of nanomaterials that have garnered significant interest in the biomedical field due to their customizable optical properties and high tissue compatibility. Since their introduction, a variety of QDs with distinct advantages have been developed to meet various clinical needs in biomedicine. Starting from the initial CdSe QDs, the field has seen the gradual emergence of carbon QDs, graphene QDs, and black phosphorus QDs. These advancements have the potential to revolutionize cancer treatment and early detection imaging. The anti-tumor effects of QDs primarily involve PDT and PTT, with many QDs capable of inducing reactive oxygen species (ROS) through the generation of singlet oxygen via triplet energy transfer (TET) and enhancing the photoluminescence (PL) of conventional photosensitizers through Förster resonance energy transfer (FRET). Furthermore, QDs can be further functionalized with common modifications such as polysaccharides, lipids, proteins, organic polymers, and other modifications to achieve specific tumor targeting for effective killing and cellular imaging applications, thereby minimizing undesired side effects.
.
Changes in the text: We have modified our text as advised (see Page 18, Lines 674-687)
Reviewer 3 Report
Comments and Suggestions for Authors
The manuscript proposes a very interesting review recap on quantum dots for theranostic and multifunctions.
The topic correlates to the journal.
The work has a clear structure.
All sections are required for a complete understanding.
Nevertheless, there are some minor issues that require to be addressed before proceeding with the publication, to enhance the quality and presentation to a broad audience.
Although catchy, the title would benefit a little remodelling: it is worthy to point out that it is not correct to address any of such construct as "biomaterial", as they are not biomaterials. Moreover, the title should be shortened.
An English check would strongly boost the whole manuscript. Check thoroughly for typos.
Author Response
Comment 1: The topic correlates to the journal. The work has a clear structure. All sections are required for a complete understanding. Nevertheless, there are some minor issues that require to be addressed before proceeding with the publication, to enhance the quality and presentation to a broad audience. Although catchy, the title would benefit a little remodeling: it is worthy to point out that it is not correct to address any of such construct as "biomaterial", as they are not biomaterials. Moreover, the title should be shortened.
Reply 1: Thank you very much for your suggestion. The title of the article has been changed to “Quantum Dot as a Potential Multifunctional material for the Enhancement of Strategies for the Clinical Diagnosis and Treatment of Cancer”
Changes in the text: We have modified our text as advised (see Page 1, Lines 1-3)
Reviewer 4 Report
Comments and Suggestions for Authors
The review entitled “Quantum Dot as a Potential Multifunctional Biomaterial Enhances the Clinical Diagnostic Ability and Therapeutic Effect of Cancer” by Wenqi Guo, Xueru Song, Jiaqi Liu, Wanyi Liu, Xiaoyuan Chu, Zengjie Lei is focused on the potential application of Quantum Dots (QD) for cancer in vitro and in vivo imaging and its therapy. Some modification of QD in particular with organic ligands and macromolecules are considered. I believe that the goals and conclusions of the review satisfied the aims of Nanomaterials and it can be accepted for publication after revision.
1. I recommend adding a chapter with consideration of basic synthesis techniques of different types of QDs.
2. QDs should be carefully considered from the point of view of cytotoxicity and biodistribution.
3. Please clarify, how are quantum dots eliminated from the body?
4. Please clarify, risks of QDs medical applications.
5. “4. The application of QDs in tumor targeted therapy” are there any study of medical applications of QDs? If it is a small animal model only the chapter should be entitled as a potential application or like that.
Author Response
Thank you for reviewing our manuscript and providing constructive feedback, which greatly assisted in enhancing the clarity and flow of the review. We have carefully revised the manuscript and addressed each point in the response provided below. The revised sections are highlighted in yellow in the manuscript, while our responses are presented in blue text.
Comment 1: I recommend adding a chapter with consideration of basic synthesis techniques of different types of QDs.
Reply 1: Thank you very much for your valuable suggestions. I have added common synthesis methods of QDs in Chapter 2 and used tables to list the common synthesis methods of QDs more clearly (from top to bottom and bottom to top, please see the table in the attachment.).
QDs can be synthesized through physical, chemical, and biological processes. Different synthesis methods will produce QDs with different luminescent properties and biocompatibility. The synthesis methods of QDs can be categorized into two main types: top-down and bottom-up. Top-down synthesis involves reducing the volume of large bulk semiconductors to create QDs, while bottom-up synthesis typically involves self-assembly. Each of these methods has its own set of advantages and disadvantages.
Changes in the text: We have modified our text as advised (see Page 3, Lines 113-118 and table 1)
Comment 2: QDs should be carefully considered from the point of view of cytotoxicity and biodistribution.
Reply 2: For practical biological applications of materials, the biological toxicity of QDs cannot be ignored. QDs mainly produce biological toxicity through chemical composition, particle size, different surface modifications, exposure routes, exposure time and dosage. In the final discussion chapter of this article, further discussion on the biological toxicity of QDs is added.
Fewer studies have focused on exploring the long-term toxicity of different QDs on living organisms. The toxicity of QDs is primarily determined by factors including chemical composition, particle size, surface modifications, exposure pathways, duration of exposure, and dosage.[165]. i.) The toxicity caused by chemical composition is mainly due to the heavy metal core of metal QDs. Such QDs may be ingested into the cytoplasm through cellular endocytosis, and prolonged presence in the cellular environment may damage the outer coating of the QDs, releasing the heavy metal core of the QDs[166]. ii.) The surface reactivity of nanoscale QDs raises toxicological concerns, which are often addressed through surface passivation techniques such as polyethylene glycol. QDs are semiconductor materials typically smaller than 20nm, making them easily absorbed by the human body and distributed in various organs. Research indicates that larger QDs tend to exhibit higher toxicity levels[167] (Accumulation of Spleen). iii.) Surface modification is typically done to enhance biocompatibility; however, improper modification techniques can introduce unknown toxicity. Rashi et al. studied the toxicity of CdSe QDs with various surface modifications on human bronchial epithelial cells. Their research revealed that the charge of the QDs has a more significant effect on toxicity compared to the size of functionalization [168]. iv.) Normally, QDs are injected intravenously into the body, which can potentially harm blood vessels [169]. But there are also studies indicating that due to the small size of QDs, they can also enter the body through the skin and respiratory tract [170]. v.) The toxicity of QDs is intricately linked to the dosage and duration of exposure. Higher concentrations and longer exposure times of QDs tend to increase their biological toxicity. In a study by Wang et al., it was observed that exposing mouse liver and kidneys to CdTe QDs led to a dose-dependent elevation in the activities of superoxide dismutase (SOD), catalase (CAT), and glutathione peroxidase (GPx) [171].
Changes in the text: We have modified our text as advised (see Page 19, Lines 688-711)
Comment 3: Please clarify, how are quantum dots eliminated from the body?
Reply 3: There is currently insufficient clinical data to support the actual metabolism of QDs in the body. However, in vivo experiments in mice and zebrafish further explored the in vivo elimination methods of QDs. Whether QDs can be cleared depends on the particle size.
In addition to considering the biological toxicity of QDs, it is crucial to assess their potential accumulation in the body over time and how the human body eliminates these nanoparticles. Some studies suggest that QDs with a diameter below 5.5 nm can be excreted through urine, while those larger than 8 nm tend to persist in the bloodstream [172]. Ge et al. also found that AgSe QDs (<3 nm) could be removed from mouse kidneys after 168 h without long-term accumulation [173]. In addition to renal clearance, QDs can bind to serum proteins and be excreted through the liver [174]. The biological toxicity of QDs is influenced by various factors, making it necessary for the biological safety assessment of QDs to consider multiple aspects rather than simply labeling them as 'toxic' or 'non-toxic'. More clinical data is required to understand the metabolism and excretion of QDs in humans. The introduction of new inorganic QDs like carbon-based QDs (CQDs), graphene QDs (GQDs), black phosphorus QDs (BPQDs), and perovskite QDs (PQDs) is expected to reduce the biological toxicity associated with older QDs. For instance, studies have demonstrated that GQDs exhibit low toxicity, and further surface modifications can enhance their therapeutic effects while ensuring long-term stability during biological therapy.
Changes in the text: We have modified our text as advised (see Page 19, Lines 711-726)
Comment 4: Please clarify, risks of QDs medical applications
Reply 4: The risks of QDs medical applications are closely related to their biological toxicity and elimination methods in the body. Currently, there are no completely non-toxic QDs, and there is not enough clinical data to explore the long-term metabolism and toxicity of QDs in the human body. Change. Therefore, using more inorganic QDs and improving their targeting and reducing off-target effects may be an effective way to reduce the risk of QDs.
More clinical data is required to understand the metabolism and excretion of QDs in humans. The introduction of new inorganic QDs like carbon-based QDs (CQDs), graphene QDs (GQDs), black phosphorus QDs (BPQDs), and perovskite QDs (PQDs) is expected to reduce the biological toxicity associated with older QDs. For instance, studies have demonstrated that GQDs exhibit low toxicity, and further surface modifications can enhance their therapeutic effects while ensuring long-term stability during biological therapy. Novel nanomaterials like black phosphorus QDs show promise in lowering biotoxicity risks due to phosphorus being a natural element in the human body. These advancements aim to enhance the targeting efficiency of QDs, minimize off-target effects, improve QD accumulation in tumor tissues, and diminish impacts on normal tissues. The ongoing advancement of quantum dots (QDs) holds promise for creating materials with superior optical properties, reduced toxicity, and enhanced biocompatibility. These advancements have the potential to enhance the efficacy of various quantum dot-based therapies in cancer treatment, including loading chemotherapy drugs onto QDs, utilizing QDs for photothermal therapy (PTT) and photodynamic therapy (PDT), and exploring gene therapies. Furthermore, the continued development of QDs is expected to lead to the creation of improved imaging agents, bolster bioimaging technology, and enable precise in vivo imaging of cancer cells with deep tissue penetration and minimal background fluorescence.
Changes in the text: We have modified our text as advised (see Page 19, Lines 720-734)
Comment 5: “4. The application of QDs in tumor targeted therapy” are there any study of medical applications of QDs? If it is a small animal model only the chapter should be entitled as a potential application or like that.
Reply 5: Thank you very much for your suggestion. The title of the article has been changed to “The potential application of QDs in tumor targeted therapy”
Changes in the text: We have modified our text as advised (see Page 10, Lines 364)

Round 2
Reviewer 4 Report
Comments and Suggestions for Authors
Dear Authors,
Thank you for your responses. I think that the manuscript can be accept in present form